# Accuracy of Vegetation Indices in Assessing Different Grades of Grassland Desertification from UAV

**DOI:** 10.3390/ijerph192416793

**Published:** 2022-12-14

**Authors:** Xue Xu, Luyao Liu, Peng Han, Xiaoqian Gong, Qing Zhang

**Affiliations:** 1Ministry of Education Key Laboratory of Ecology and Resource Use of the Mongolian Plateau, School of Ecology and Environment, Inner Mongolia University, Hohhot 010021, China; 2Collaborative Innovation Center for Grassland Ecological Security (Jointly Supported by the Ministry of Education of China and Inner Mongolia Autonomous Region), Hohhot 010021, China

**Keywords:** grassland desertification, vegetation index, fractional vegetation coverage, UAV visible light images, the Mu Us Sandy

## Abstract

Grassland desertification has become one of the most serious environmental problems in the world. Grasslands are the focus of desertification research because of their ecological vulnerability. Their application on different grassland desertification grades remains limited. Therefore, in this study, 19 vegetation indices were calculated for 30 unmanned aerial vehicle (UAV) visible light images at five grades of grassland desertification in the Mu Us Sandy. Fractional Vegetation Coverage (FVC) with high accuracy was obtained through Support Vector Machine (SVM) classification, and the results were used as the reference values. Based on the FVC, the grassland desertification grades were divided into five grades: severe (FVC < 5%), high (FVC: 5–20%), moderate (FVC: 21–50%), slight (FVC: 51–70%), and non-desertification (FVC: 71–100%). The accuracy of the vegetation indices was assessed by the overall accuracy (OA), the kappa coefficient (*k*), and the relative error (RE). Our result showed that the accuracy of SVM-supervised classification was high in assessing each grassland desertification grade. Excess Green Red Blue Difference Index (EGRBDI), Visible Band Modified Soil Adjusted Vegetation Index (V-MSAVI), Green Leaf Index (GLI), Color Index of Vegetation Vegetative (CIVE), Red Green Blue Vegetation Index (RGBVI), and Excess Green (EXG) accurately assessed grassland desertification at severe, high, moderate, and slight grades. In addition, the Red Green Ratio Index (RGRI) and Combined 2 (COM_2_) were accurate in assessing severe desertification. The assessment of the 19 indices of the non-desertification grade had low accuracy. Moreover, our result showed that the accuracy of SVM-supervised classification was high in assessing each grassland desertification grade. This study emphasizes that the applicability of the vegetation indices varies with the degree of grassland desertification and hopes to provide scientific guidance for a more accurate grassland desertification assessment.

## 1. Introduction

Grassland desertification is a serious ecological and environmental problem in arid and semi-arid areas [1]. Thus, how to effectively mitigate and combat grassland desertification has become the focus of global ecological and environmental studies [2]. China has nearly 400 million hectares of natural grasslands, accounting for 41.7% of the country’s land area. However, 90% of these grasslands show different degrees of degradation [3]. Grassland desertification is one of the most important aspects of grassland degradation [4]. Grassland desertification seriously affects the balance of ecosystems in a region and leads to the deterioration of the ecological environment, such as local soil erosion and fertility decline [5]. Therefore, assessing the degree of grassland desertification plays an important role in grassland restoration and continued healthy development.

Grassland desertification assessment is important for effective desertification management [6]. Remote sensing is the main method used to assess the extent of desertification in large-scale grasslands [7]. Initially, scholars used empirical visual interpretation to assess desertification, but this is usually labor-intensive and time-consuming and is not suitable for assessing large areas [8]. Moreover, the results of visual interpretation can be very inaccurate [9]. The development of remote sensing indices has facilitated desertification assessment, and a single remote sensing index is widely used in research [10,11,12,13]. However, since a single remote sensing index is greatly influenced by atmospheric and soil factors, its assessment results have certain limitations. For example, the Normalized Difference Vegetation Index (NDVI) is related to humidity, and the Enhanced Vegetation Index (EVI) is affected by soil moisture [14]. It is more reasonable to combine a single remote sensing index with various other remote sensing data, such as precipitation and soil data, to monitor desertification [15].

Fractional Vegetation Coverage (FVC) and its changes can directly reflect the changes in the ecological environment in desertification areas and indicate their climate, hydrology, and ecology [16]. Thus, FVC is an important indicator that reflects the degree of desertification [17]. Traditional vegetation cover can be obtained by field investigation, but the process takes up considerable labor and economic resources [18]. The supervised classification method consumes considerable storage space and time, so its usage is limited in large-scale research [19]. With the continuous optimization of the vegetation index, the calculation of the vegetation index through remote sensing images can more quickly and effectively reflect the vegetation coverage at any scale on the surface [10,20,21]. Therefore, methods based on vegetation indices have been used for estimating vegetation cover [20].

Some vegetation indices are constructed based on ratios such as the NDVI, Soil-Adjusted Vegetation Index (SAVI), Modified SAVI (MSAVI), Green Red Vegetation Index (GRVI), and EVI [20,22,23,24,25,26]. Some vegetation indices are constructed in the form of linear combinations, such as the Perpendicular Vegetation Index (PVI), Weighted Difference Vegetation Index (WDVI), Tasseled Cap Greenness (TCG), and Transformed Vegetation Index (TVI) [27,28,29]. Some vegetation indices are constructed based on soil information, such as Soil-Adjusted and Atmospherically Resistant Vegetation Index (SARVI) and Modified Non-Linear Vegetation Index (MNLI) [30,31]. A series of vegetation indices have been shown to be closely related to desertification and can quickly assess regional desertification [32]. Thus, this study assessed the applicability of these vegetation indices in desertification to provide a scientific basis for grassland desertification assessment [33].

Some studies have found that the applicability of vegetation indices is usually related to datasets, sensors, soil type, atmosphere, etc. These factors have led to a large variation in the applicability of vegetation indices [30,34]. Although many studies have conducted grassland desertification assessments using vegetation indices, to our knowledge, no research has explored the application of vegetation indices to different grades of grassland desertification. Usually, desertification is classified into five levels, which are severe, severe, moderate, slight, and non-desertification. The lower the FVC, the more serious the desertification. With the advantages of low cost, simple operation, and high spatial resolution, UAVs have been widely used in many fields, such as vegetation monitoring and environmental assessment [35,36,37,38]. However, a few studies have conducted desertification assessments based on UAV visible light images [39].

In this study, we selected 19 vegetation indices for evaluating grassland desertification based on UAV visible light images in a typical area of the Mu Us Sandy. We assessed the applicability of the vegetation indices at different grades of grassland desertification. The objectives of this study were to calculate visible vegetation indices using UAV images, to assess the accuracy of vegetation indices under five grassland desertification grades, and to find the vegetation indices suitable for assessing desertification in each grade.

## 2. Materials and Methods

### 2.1. Study Area

The study area is located in a typical area of the Mu Us Sandy (108°17′–109°40′ E, 37°38′–39°23′ N), which has a temperate continental monsoon climate with a mean annual temperature of 6–9 °C. It has an increase in the mean annual precipitation from approximately 250 mm in the northwest to approximately 440 mm in the southeast. The altitude of the area is 1200–1600 m, and the terrain slopes from northwest to southeast [40,41]. The main soils are Kastanozems, Arenosols, Histosols, and Solonchaks. The zonal vegetation in the study area is dominated by *Stipa bungeana* and *Thymus serpyllum* [42].

### 2.2. UAV Visible Light Image Acquisition

This study was conducted during the vegetation growing season from 28 August to 5 September 2021. Images are taken from 11 a.m. to 3 p.m. daily. During this period, the weather was good when the images were taken. The average wind speed of the images taken was 8 km/h, the average visibility was 25 km, and the average relative humidity was 31%. Thus, the obtained UAV images were less affected by meteorological factors and other factors. The DJI Air 2s drone (with a camera of 20 megapixels and a 1-inch image sensor) was used to take pictures in different areas with large differences in grassland desertification grades (Figure 1). The imagery produced by the consumer-grade visible light UAV provides an economical and efficient method for monitoring and evaluating the surface vegetation, so the UAV was selected for grassland desertification assessment. The UAV images were taken with the camera pointing vertically downward at a flight height of 10 m and a spatial resolution of approximately 0.20 cm/pixel. To obtain a clearer image and eliminate the distortion of the image as much as possible, we chose a flying height of 10 m. Because this study does not involve the central wavelength position of each band and the range of the band, radiometric correction of the obtained UAV images is not needed [43]. DJI Air 2s drone does not support waypoint flight, so a single image was used.

### 2.3. Supervised Classification and Desertification Grades

In this paper, we conducted 15 flights with a total of 15 sites (Figure 1), and 2 images were randomly selected for each site. A total of 30 images of different grassland desertification grades were selected. According to the previous studies [44,45], this paper selects 30 images to meet the statistical minimum sample size. We selected 60 vegetation and 60 non-vegetated regions of interest (ROIs) as the training set from each image for supervised classification by support vector machine (SVM) to obtain the desertification grades from these images. The kernel function is a radial basis function, the gamma in the kernel function is 0.333, and the classification probability threshold is a default value of 0. For each image, we selected 15 vegetated and 15 non-vegetated ROIs as the test set. We calculated the confusion matrix to derive the overall accuracy (OA) and the kappa coefficient (*k*) to verify the accuracy of the supervised classification results. We counted each image’s vegetation and non-vegetation cells after supervised classification to obtain the FVC and used it as the reference value. The above process was calculated based on ENVI 5.3 software [46,47].

The confusion matrix is an effective descriptive tool to demonstrate accuracy assessment, and it is the most basic, intuitive, and computationally simplest method for measuring classification accuracy [48]. Therefore, we chose the confusion matrix to evaluate the classification accuracy. OA is one of the most common evaluation metrics that can be used to assess a model’s performance visually. The higher the accuracy, the better the classifier. *k* is applied to determine whether the model’s results are consistent with the actual results: *k* = 1 indicates that the results are completely consistent; *k* ≥ 0.75 is considered satisfactory; and *k* < 0.4 is considered not ideal [49].
(1)$$OA=\frac{TP+TN}{TP+FN+FP+TN}$$
(2)$$k=\frac{N\sum_{i=1}^{r}x_{ii}-\sum_{i=1}^{r}\left(x_{i+}x_{+i}\right)}{N^{2}-\sum_{i=1}^{r}\left(x_{i}{}_{+}x_{+i}\right)}$$
where the number of pixels correctly classified as positive samples is denoted by *TP*, and the number of pixels correctly classified as negative samples is denoted by *FN*. The number of pixels with errors for negative samples is denoted by *FP*, and the number of pixels with errors for positive samples is denoted by *TN*. In the equation of *k*, *r* is the total number of categories in the confusion matrix; *N* is the total number of pixels used for accuracy evaluation; $x_{ii}$ is the total number of pixels correctly extracted in the confusion matrix; $x_{i+}$ and $x_{+i}$ are the total number of pixels for each row and column of the confusion matrix.

According to the classification standard of desertification grades used in the 1:100,000 Distribution Atlas of Chinese Deserts published by the “Environmental & Ecological Science Data Center for West China, National Natural Science Foundation of China (http://westdc.westgis.ac.cn) and China’s national standard Technical Code of Practice on the Sandified Land Monitoring (GB/T 24255-2009). Desertification is classified as severe desertification, heavy desertification, moderate desertification, light desertification, and non-desertification based on the FVC from the supervised classification (Table 1) [50].

### 2.4. Vegetation Index Assessment of Desertification

To assess grassland desertification, we selected 19 vegetation indices to calculate the FVC of UAV visible light images (Table 2). Vegetation and soil were separated by vegetation indices, and vegetation and non-vegetation areas in the image were segmented by the Otsu thresholding method to obtain a binary image [51]. Then, the numbers of vegetation and non-vegetation cells were counted to obtain the FVC of the image. This method used a threshold to separate the image into two parts, the target and background (vegetation and non-vegetation), based on the gray-level distribution of the image. When the between-class variance target and background are larger, the difference between these two parts is larger. When part of the target is misclassified into the background or part of the background is misclassified into the target, the between-class variance decreases. The maximum between-class variance can minimize the misclassification and omission in the segmentation and then extract the target information effectively [52]. The method has better results for images where the target and background have distinctly different grayscale features. The calculation is simple and fast and is not affected by image brightness and contrast [53]. Therefore, this study selected this method to segment the vegetation and non-vegetation areas on the image. The above calculations were performed using Python 3.7.

### 2.5. Accuracy Verification and Statistical Analysis

In this study, we used two methods to evaluate the accuracy of the vegetation indices for assessing grassland desertification. First, 80 vegetated and 80 non-vegetated surface realistic ROIs were randomly selected as references from each image, and the confusion matrix was calculated to derive the OA and *k* [46]. They did not overlap with the train and test set above, and contain approximately 12% of the entire image area.

Assessment of the accuracy of vegetation indices based on surface realistic ROI takes advantage of the ease of visual identification of vegetation and non-vegetation information in high-resolution images [66]. Although the vegetation and non-vegetation realistic areas of interest are selected evenly over the entire image, they are still partial validations of the image elements, and human error cannot be avoided [63]. This study also chose RE as another accuracy evaluation method to reduce this error. The supervised classification results of the UAV visible light images were taken as the reference values of surface vegetation cover, and the Relative Error (RE) of the FVC by vegetation indices was calculated. The smaller the RE is, the higher the accuracy of the vegetation indices is [67]. The average value of the accuracy verification from 6 images of every desertification grade was calculated and used as the final result of different grassland desertification grades.
(3)$$RE=\frac{\left|V_{SUP}-V_{VI}\right|}{V_{SUP}}$$
where RE is the relative error of FVC, $V_{SUP}$ is the FVC obtained by supervised classification, and $V_{VI}$ is the FVC obtained by the vegetation index.

To assess the applicability of the vegetation indices under different grades of grassland desertification, all statistical analyses in this study were performed by one-way analysis of variance (ANOVA) using Python 3.7. Duncan’s method was used to test for significant differences between the three accuracy verification indicators of different vegetation indices in five desertification grades (*p* < 0.05) [60].

This study only distinguished vegetation and non-vegetation areas. The images taken by the UAV had a high resolution of 0.2 cm, which was able to monitor most of the vegetation features. Therefore, the validation samples were directly selected on the image without field measurements.

## 3. Results

### 3.1. UAV Visible Light Image Surveillance Classification

The OA of the supervised classification for all 30 images was above 98%, and the k was above 0.96, so the supervised classification results were accurate (Table 3). Therefore, the FVC calculated after the supervised classification was taken as the true value of the image. Based on the FVC of 30 images, five grades of grassland desertification were classified as severe desertification, severe desertification, moderate desertification, mild desertification, and non-desertification.

### 3.2. Visible Light Vegetation Index Accuracy Assessment

#### 3.2.1. Vegetation Index Accuracy Assessment of the Severe Desertification Grade

There were significant differences between the 19 vegetation indices under the three accuracy indicators of the severe desertification grade (Table 4). The accuracies of EGRBDI, V-MSAVI, GLI, RGBVI, CIVE, EXG, COM_2_, NGBDI, EXB, RGRI, GBRI, and VEG were significantly higher than those of COM, DEVI, and g in the OA (Figure 2A). The accuracies of V-MSAVI, EGRBDI, GLI, RGBVI, CIVE, EXG, RGRI, and COM_2_ in the *k* were significantly higher than those of EXR, COM, g, and DEVI (Figure 2B). The accuracies of the remaining indices in the RE were significantly higher than those of DEVI, g, and COM (Figure 2C). Therefore, V-MSAVI, EGRBDI, GLI, RGBVI, CIVE, EXG, RGRI, and COM_2_ were more accurate and stable for severe desertification grade assessment.

#### 3.2.2. Vegetation Index Accuracy Assessment of the High Desertification Grade

There were significant differences between the 19 vegetation indices under the three accuracy indicators of the high desertification grade (Table 5). The accuracies of EGRBDI, V-MSAVI, CIVE, GLI, RGBVI, and EXG were significantly higher than those of DEVI, EXR, GBRI, and COM in the OA (Figure 3A). The accuracies of EGRBDI, V-MSAVI, GLI, CIVE, RGBVI, EXG, and COM_2_ in the *k* were significantly higher than those of g, DEVI, EXR, GBRI, and COM (Figure 3B). The accuracies of the remaining indices in the RE were significantly higher than those of DEVI and COM (Figure 3C). Therefore, EGRBDI, V-MSAVI, GLI, CIVE, RGBVI, and EXG were more accurate and stable for high desertification grade assessment.

#### 3.2.3. Vegetation Index Accuracy Assessment of the Moderate Desertification Grade

There were significant differences between the 19 indices under the three accuracy indicators of the high desertification grade (Table 6). The accuracies of the remaining indices were significantly higher than those of DEVI, g, and GBRI in the OA (Figure 4A). The accuracies of EGRBDI, GLI, V-MSAVI, CIVE, RGBVI, and EXG in the *k* were significantly higher than those of DEVI, g, and GBRI (Figure 4B). The accuracies of RGBVI, GLI, EXGR, EXR, NGRDI, MGRVI, COM, V-MSAVI, EGRBDI, and VEG in the RE were significantly higher than those of the other indices (Figure 4C). Therefore, EGRBDI, GLI, V-MSAVI, CIVE, RGBVI, and EXG were more accurate and stable for moderate desertification grade assessment.

#### 3.2.4. Vegetation Index Accuracy Assessment of the Slight Desertification Grade

There were significant differences between the 19 indices under the three accuracy indicators of the high desertification grade (Table 7). The accuracies of EGRBDI, V-MSAVI, RGBVI, GLI, CIVE, and EXG were significantly higher than those of GBRI, COM_2_, COM, DEVI, and g in the OA (Figure 5A). The accuracies of EGRBDI, V-MSAVI, RGBVI, GLI, CIVE, EXG, VEG, NGBDI, and EXB in the *k* were significantly higher than those of GBRI, COM_2_, COM, DEVI, and g (Figure 5B). g was significantly better than COM_2_ in the RE, but the OA of g and *k* was lower (Figure 5C). Therefore, EGRBDI, V-MSAVI, RGBVI, GLI, CIVE, and EXG were more accurate and stable for slight desertification grade assessment.

#### 3.2.5. Vegetation Index Accuracy Assessment of the Non-Desertification Grade

Significant differences between the 19 vegetation indices in the *k* at the non-desertification grade and non-significant differences between the 19 vegetation indices in the OA and RE were observed (Table 8). The accuracies of EGRBDI, RGBVI, GLI, EXG, RGRI, V-MSAVI, CIVE, NGRDI, MGRVI, EXGR, EXR, and GBRI in the *k* were significantly higher than those of the other indices (Figure 6B). The RE of VEG and DEVI was smaller (Figure 6C), but their OA and *k* were lower (Figure 6A,B), so the accuracy of VEG and DEVI were lower. The FVC derived from the vegetation indices was similar to the supervised classification values, but the OA and *k* for all indices were low. There were misclassifications of images, and 19 vegetation indices had low accuracy. All vegetation indices were not applicable to assessments at the non-desertification grade.

## 4. Discussion

Previous studies have shown that supervised classification is one of the tools for the effective identification of vegetation information [68,69]. In this study, SVM supervised classification of 30 UAV visible light images revealed that the method was accurate in assessing grassland desertification (Table 1). Therefore, using the FVC obtained by SVM supervised classification as reference values is reliable for dividing desertification grades. Ma et al. showed that SVM is accurate in planting structures of varying complexity, which is consistent with our results [70]. The reason is that SVM is based on the Vapnik–Chervonenkis dimension theory of statistical learning and the principle of minimum structural risk. SVM provides a good balance between the complexity of the model and the learning ability [70]. In addition, SVM transforms a complex learning problem into a high-dimensional simplified linear problem, which can improve reliability and control classification ability [70,71].

Our study found that V-MSAVI, EGRBDI, GLI, RGBVI, CIVE, and EXG are suitable for the assessment of severe, high, and moderate desertification. RGRI and COM_2_ are suitable for the assessment of slight desertification. In addition, there were no vegetation indices suitable for desertification assessment of the non-desertification grade. This indicated that the application of vegetation indices was not the same in different desertification grades. Our findings are similar to the results of Lima-Cueto et al., who used vegetation indices to quantify olive grove cover [72]. Their study indicated that the vegetation indices were affected by olive grove coverage and that the vegetation indices were less accurate at high coverage. This is mainly because the ability of the vegetation indices to distinguish between vegetation and soil is weakened in areas with very high coverage heterogeneity. Meanwhile, our results are similar to those of Zhao et al., who extracted maize vegetation cover based on visible light UAV images [73]. They concluded that maize cover affects the accuracy of the vegetation indices and that the accuracy of the vegetation indices decreases with increasing maize cover. The main reason is that vegetation leaves reflect light under the sunlight, and the increase in the area of the shaded part with the increase in vegetation cover.

The accuracy of the vegetation index varies in different classes of grassland desertification, which may be due to the following reasons. In the visible light range, healthy green vegetation has a strong reflection in the green light band and strong absorption in the blue and red light bands [74]. V-MSAVI, EGRBDI, GLI, RGBVI, and EXG use the square of the green band or two times the green band to further enhance the strong reflection of vegetation in the green band, improving the ability of the index to identify green vegetation information. Determined by the spectral reflection characteristics of vegetation and soil, there is usually a spectral overlap between vegetation and bare soil in the red and green bands [43]. Therefore, vegetation indices from a single band and those containing only green-red light bands or red-blue light bands do not separate vegetated and non-vegetated areas well [75]. Both CIVE and the above five vegetation indices utilize the combined operation of red, green, and blue bands to extract vegetation information effectively. The accuracy of the RGRI and COM_2_ indices is high in severe desertification. This may be because there is very little green vegetation in severe desertification, and most of them are bare soil, which weakens the influence of vegetation and soil spectral overlap on the extraction of vegetation information by these two indices. These factors affect the accuracy and lead to difficulty identifying vegetation in desert grassland areas. Therefore, the six vegetation indices can be well used for desertification assessment compared to other vegetation indices.

Our study found that the vegetation index is not applicable to the desertification assessment of the non-desertification grade. This is mainly because UAV photography was taken during the vegetation bloom period, and vegetation had a very high coverage in some areas. High vegetation coverage often means high spatial heterogeneity [76]. Generally, the greenness of desert grassland vegetation is less than that of other grassland types, and some vegetation appears brown or yellowish brown. The color of desert grassland non-vegetation is closer to yellow, and it is difficult to distinguish vegetation and non-vegetation parts from each other at the color level [74]. Sufficient remote sensing information is needed to reflect vegetation characteristics. Therefore, it is difficult to distinguish vegetation and non-vegetation using only the vegetation index constructed only from visible light [77]. This study empirically demonstrated that supervised classification methods were more effective in grassland desertification assessment of the non-desertification grade (Figure 6). In the future, new methods, such as near-infrared and visible band combinations, color mixture analysis, image texturing, and neural network machine learning, will be helpful to improve the accuracy of grassland desertification assessment [55,78].

## 5. Conclusions

Our study collected UAV visible light images of different grades of desertification grassland during the plant growing season. We evaluated the accuracy of 19 vegetation indices in 5 grades of grassland desertification. V-MSAVI, EGRBDI, GLI, RGBVI, CIVE, and EXG have high accuracy in assessing severe, high, and moderate-grade desertification. RGRI and COM_2_ have high accuracy in assessing slight-grade desertification. All vegetation indices have low accuracy in non-desertification. This study emphasized that the application of vegetation indices was controlled by the grade of desertification. Therefore, we suggest that the desertification grade should be considered when using the vegetation indices to assess grassland desertification.

## Figures and Tables

**Figure 1 ijerph-19-16793-f001:**
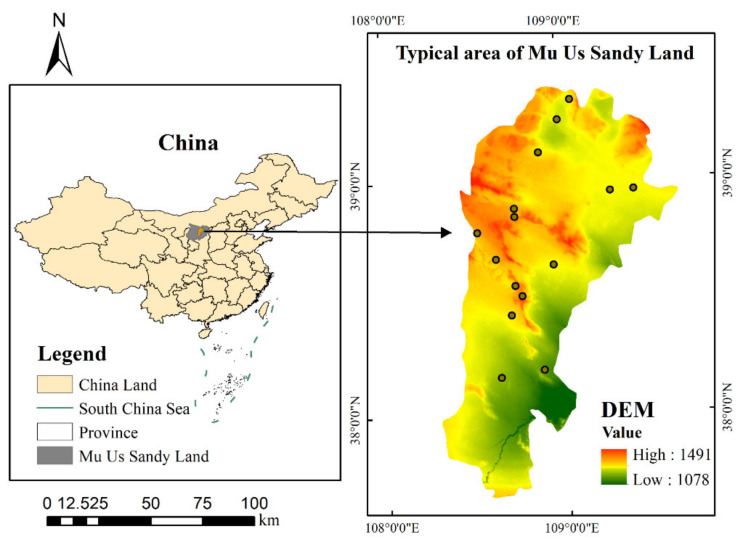
Location of the study area.

**Figure 2 ijerph-19-16793-f002:**
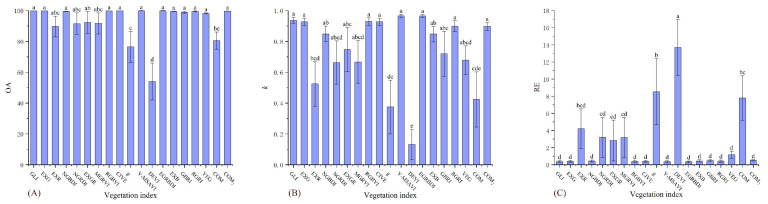
OA, *k*, and RE differences of 19 vegetation indices on the severe desertification grade. (**A**) figure is the result of one-way analysis of variance for OA. (**B**) figure is the result of one-way analysis of variance for *k*. (**C**) figure is the result of one-way analysis of variance for RE. Values with the same lowercase letters within vegetation indices are not significantly different at *p* < 0.05.

**Figure 3 ijerph-19-16793-f003:**
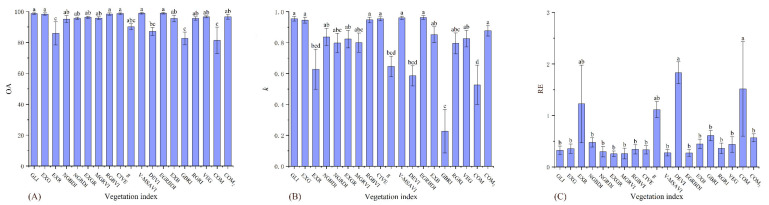
OA, *k*, and RE differences of 19 vegetation indices on the high desertification grade. (**A**) figure is the result of one-way analysis of variance for OA. (**B**) figure is the result of one-way analysis of variance for *k*. (**C**) figure is the result of one-way analysis of variance for RE. Values with the same lowercase letters within vegetation indices are not significantly different at *p* < 0.05.

**Figure 4 ijerph-19-16793-f004:**
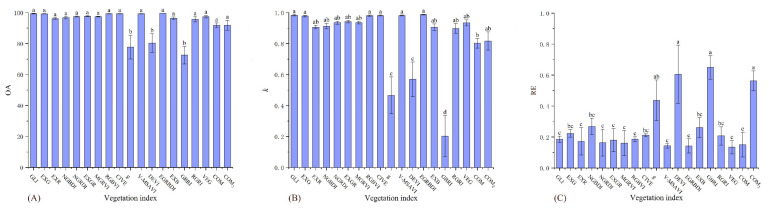
OA, *k*, and RE differences of 19 vegetation indices on the moderate desertification grade. (**A**) figure is the result of one-way analysis of variance for OA. (**B**) figure is the result of one-way analysis of variance for *k*. (**C**) figure is the result of one-way analysis of variance for RE. Values with the same lowercase letters within vegetation indices are not significantly different at *p* < 0.05.

**Figure 5 ijerph-19-16793-f005:**
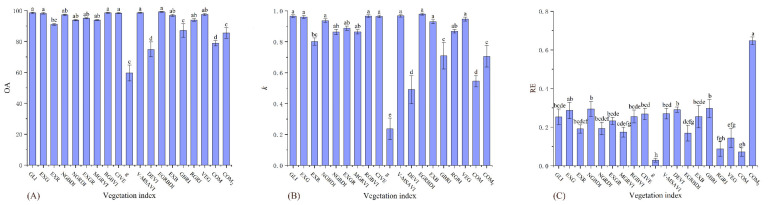
OA, *k*, and RE differences of 19 vegetation indices on the slight desertification grade. (**A**) figure is the result of one-way analysis of variance for OA. (**B**) figure is the result of one-way analysis of variance for *k*. (**C**) figure is the result of one-way analysis of variance for RE. Values with the same lowercase letters within vegetation indices are not significantly different at *p* < 0.05.

**Figure 6 ijerph-19-16793-f006:**
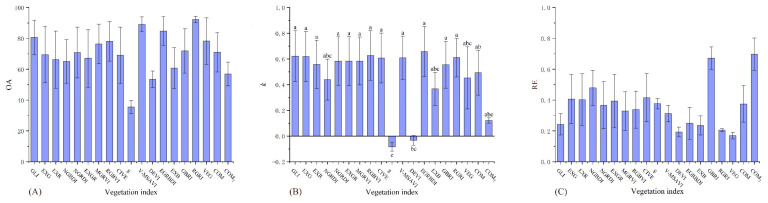
OA, *k*, and RE differences of 19 vegetation indices on the non-desertification grade. (**A**) figure is the result of one-way analysis of variance for OA. (**B**) figure is the result of one-way analysis of variance for *k*. (**C**) figure is the result of one-way analysis of variance for RE. Values with the same lowercase letters within vegetation indices are not significantly different at *p* < 0.05.

**Table 1 ijerph-19-16793-t001:** Desertification grades.

Desertification Grades	FVC	Desertification Area in the Image	UAV Visible Light Images
Severe	<5%	≥95 m^2^	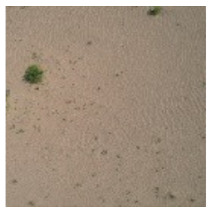
High	5–20%	80–94 m^2^	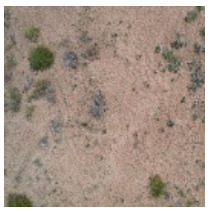
Moderate	21–50%	50–79 m^2^	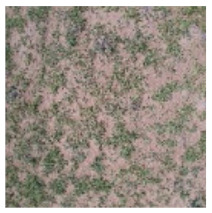
Slight	51–70%	30–49 m^2^	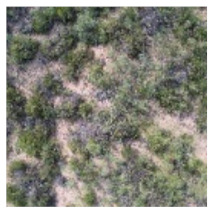
Non-desertification	>70%	<30 m^2^	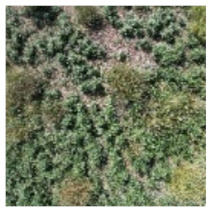

Note: The sample images are all from the 30 images in this study.

**Table 2 ijerph-19-16793-t002:** Vegetation indices.

Vegetation Index	Full Name	Equation
GLI [54]	Green Leaf Index	(2 × G − R − B)/(2 × G + R + B)
ExG [55]	Excess Green	2g − r − b
ExR [56]	Excess Red	1.4r − g
ExB [57]	Excess Blue	1.4b − g
NGBDI [56]	Normalized Green Red Difference Index	(G − B)/(G + B)
NGRDI [56]	Normalized Green Red Difference Index	(G − R)/(G + R)
ExGR [56]	Excess Green Minus Excess Red	E × G − E × R
MGRVI [58]	Modified	(G^2^ − R^2^)/(G^2^ + R^2^)
RGBVI [58]	Red Green Blue Vegetation Index	(G^2^ − B × R)/(G^2^ + B × R)
GBRI [59]	Green Blue Ratio Index	b/g
RGRI [60]	Red Green Ratio Index	r/g
CIVE [61]	Color Index of Vegetation	0.441r − 0.881g + 0.385b + 18.78745
VEG [62]	Vegetative	g/(r^α^b^1−α^)
DEVI [63]	Difference Excess Vegetation Index	G/3G + R/3G + B/3G
EGRBDI [43]	Excess Green Red Blue Difference Index	((2G)2 − B × R)/((2G)2 + B × R)
V-MSAVI [64]	Visible Band Modified Soil Adjusted Vegetation Index	$\frac{(1\;\times \;G\;+\;1)\;-\;\sqrt{{(2\;\times \;G\;+\;1)}^{2}\;-\;8\;\times \;(2\;\times \;G\;-\;R\;-\;B)}}{2}$
g [55]	Green Chromatic Coordinates	G
COM [57]	Combined	0.25E × G + 0.3E × GR + 0.33CIVE + 0.12VEG
COM_2_ [65]	Combined 2	0.36E × G + 0.47CIVE + 0.17VEG

Note: R: Red channel; G: Green channel; B: Blue channel; r: Standardization of red channel; g: Standardization of green channel; b: Standardization of blue channel. Else: $r\;=\;\frac{R}{R\;+\;G\;+\;B}$; $g\;=\;\frac{G}{R\;+\;G\;+\;B}$; $b\;=\;\frac{B}{R\;+\;G\;+\;B}$; α = 0.667.

**Table 3 ijerph-19-16793-t003:** Supervised classification results of 30 images.

Image Number	Center Coordinate		Altitude m	FVC %	Desertification Grade	OA %	*k*
	Latitude	Longitude					
1	39°20′3871″ E	109°04′4434″ N	1269.7	4.3243	Severe	99.1168	0.9821
2	38°30′3617″ E	108°04 ′0653″ N	1351.9	1.5050	Severe	99.011	0.9783
3	39°20′3910″ E	109°04′4392″ N	1352.1	4.0864	Severe	99.6525	0.9922
4	38°50′5350″ E	108°44′2955″ N	1356.0	3.3052	Severe	99.9252	0.9966
5	38°50′5348″ E	108°44′2929″ N	1356.0	3.3717	Severe	98.7145	0.9642
6	38°30′3629″ E	108°46′0643″ N	1269.9	2.8890	Severe	99.7404	0.9939
7	39°15′3308″ E	109°00′1219″ N	1267.6	13.2679	High	98.9788	0.9671
8	39°15′3364″ E	109°00′1315″ N	1267.5	18.3795	High	99.5751	0.9725
9	38°09′4113″ E	108°38′1296″ N	1247.7	6.7185	High	99.5239	0.9879
10	38°25′4054″ E	108°42′2405″ N	1293.8	16.2495	High	99.7815	0.9956
11	38°25′4082″ E	108°42′2359″ N	1293.9	15.9524	High	99.54	0.9902
12	38°09′4144″ E	108°38′1247″ N	1247.5	9.4557	High	98.8213	0.9764
13	38°38′4889″ E	108°56′4260″ N	1270.6	32.6411	Moderate	99.8111	0.9676
14	39°07′1405″ E	108°53′2209″ N	1295.8	43.2160	Moderate	99.9583	0.99
15	39°07′1527″ E	108°53′2101″ N	1296.9	31.7365	Moderate	99.5985	0.9747
16	38°33′1085″ E	108°43′5874″ N	1314.8	45.6735	Moderate	99.833	0.982
17	38°33′1040″ E	108°44′0091″ N	1314.5	31.8236	Moderate	99.6005	0.9892
18	38°38′4745″ E	108°56′4154″ N	1270.3	26.3766	Moderate	99.5415	0.9894
19	38°52′5931″ E	108°44′2805″ N	1344.1	53.0043	Slight	99.5556	0.9855
20	38°52′5931″ E	108°44′2816″ N	1344.1	55.3886	Slight	99.6876	0.9937
21	38°57′1514″ E	109°25′2166″ N	1266.0	51.5241	Slight	99.931	0.9985
22	38°57′1522″ E	109°25′2124″ N	1266.1	51.5331	Slight	99.8963	0.9928
23	38°40′7309″ E	108°37′4289″ N	1287.4	54.7885	Slight	99.6876	0.9937
24	38°40′4888″ E	108°37′4914″ N	1287.3	51.5424	Slight	99.9765	0.9946
25	38°46′4981″ E	108°31′0358″ N	1284.9	86.2322	Non-desertification	99.874	0.9606
26	38°46′5164″ E	108°31′3291″ N	1284.8	96.5132	Non-desertification	99.8873	0.9619
27	38°56′4931″ E	109°17′3098″ N	1261.4	83.4444	Non-desertification	99.4857	0.9808
26	38°56′5034″ E	109°17′2680″ N	1261.3	89.2501	Non-desertification	99.9279	0.9674
29	38°11′5005″ E	108°52′2792″ N	1266.9	71.6581	Non-desertification	99.5638	0.9833
30	38°11′5022″ E	108°52′2811″ N	1266.8	76.3327	Non-desertification	99.8111	0.9676

Note: OA is the overall accuracy. *k* is the kappa coefficient.

**Table 4 ijerph-19-16793-t004:** OA, *k*, and RE differences of 19 vegetation indices on the severe desertification grade.

		TSS	df	MS	F	*p*
OA (%)	Between Groups	14,840.249	18	824.458	5.562	0.000
	Within Groups	14,083.024	95	148.242		
	Grand Total	28,923.272	113			
*k*	Between Groups	5.895	18	0.328	5.430	0.000
	Within Groups	5.730	95	0.060		
	Grand Total	11.625	113			
RE	Between Groups	1459.716	18	81.095	4.710	0.000
	Within Groups	1635.607	95	17.217		
	Grand Total	3095.324	113			

**Table 5 ijerph-19-16793-t005:** OA, *k*, and RE differences of 19 vegetation indices on the high desertification grade.

		TSS	df	MS	F	*p*
OA (%)	Between Groups	3454.562	18	191.920	3.550	0.000
	Within Groups	5136.480	95	54.068		
	Grand Total	8591.042	113			
*k*	Between Groups	3.914	18	0.217	7.829	0.000
	Within Groups	2.639	95	0.028		
	Grand Total	6.553	113			
RE	Between Groups	23.606	18	1.311	2.584	0.002
	Within Groups	48.222	95	0.508		
	Grand Total	71.828	113			

**Table 6 ijerph-19-16793-t006:** OA, *k*, and RE differences of 19 vegetation indices on the moderate desertification grade.

		TSS	df	MS	F	*p*
OA (%)	Between Groups	6957.370	18	386.521	8.413	0.000
	Within Groups	4364.848	95	45.946		
	Grand Total	11,322.218	113			
*k*	Between Groups	4.774	18	0.265	16.451	0.000
	Within Groups	1.531	95	0.016		
	Grand Total	6.305	113			
RE	Between Groups	2.998	18	0.167	4.671	0.000
	Within Groups	3.387	95	0.036		
	Grand Total	6.385	113			

**Table 7 ijerph-19-16793-t007:** OA, *k*, and RE differences of 19 vegetation indices on the slight desertification grade.

		TSS	df	MS	F	*p*
OA (%)	Between Groups	11,608.489	18	644.916	23.374	0.000
	Within Groups	2621.216	95	27.592		
	Grand Total	14,229.705	113			
*k*	Between Groups	4.392	18	0.244	26.617	0.000
	Within Groups	0.871	95	0.009		
	Grand Total	5.262	113			
RE	Between Groups	1.785	18	0.099	13.981	0.000
	Within Groups	0.674	95	0.007		
	Grand Total	2.459	113			

**Table 8 ijerph-19-16793-t008:** OA, *k*, and RE differences of 19 vegetation indices on the non-desertification grade.

		TSS	df	MS	F	*p*
OA (%)	Between Groups	18,799.030	18	1044.391	1.009	0.457
	Within Groups	98,285.865	95	1034.588		
	Grand Total	117,084.895	113			
*k*	Between Groups	5.465	18	0.304	1.750	0.044
	Within Groups	16.485	95	0.174		
	Grand Total	21.950	113			
RE	Between Groups	2.203	18	0.122	1.692	0.054
	Within Groups	6.871	95	0.072		
	Grand Total	9.074	113			

## Data Availability

Not applicable.

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
