# Peer review of "Accuracy of Vegetation Indices in Assessing Different Grades of Grassland Desertification from UAV"

_ijerph, 2022, doi:10.3390/ijerph192416793_

Round 1

Reviewer 1 Report

The study entitled “Different accuracy of vegetation indices in assessing different grades of grassland desertification from UAV” by Xu and co-authors, overall evaluated 19 vegetation indices to assess different desertification levels in arid ecosystems of China. The authors find that several of the indices work, except for non desertification areas. Overall the study is well conducted and easy to follow. I just have (mostly) some form comments, but also a few more crucial comments. Find them all below.

Abstract

L.15. Change “the applicability of vegetation indices of different” for “their application on different”.

L.21. Add a comma (,) before “and non-desertification”.

L.23. Change “(k) and the relative” for “(k), and the relative”.

L.24. Change “OA, k and RE” for “OA, k, and RE”.

L.26. Change “RGBVI and EXG” for “RGBVI, and EXG”. It does not seem good to have these unexplained abbreviations here, though I know the space is limited.

L.28. Change “grade was low” for “grade had low”.

Introduction

L.48. Change “grassland” for “grasslands”.

L.49. Change “but it is usually” for “but this is usually”. Also, visual inspection can be very, very inaccurate.

L.59. Change “indicate the” for “indicate their”.

L.84-85. In the Introduction, you need to more specifically what you mean when you say something like “different degrees of grassland desertification”.

L.85. Change “operation and high spatial” for “operation, and high spatial”.

Materials and Methods

L.102. Change “Histosols and Solonchaks” for “Histosols, and Solonchaks”.

L.113-114. Again, explain what those desertification grades were exactly.

L.119. Change “of 10m.” for “of 10 m.”.

L.126-127. Why not more than the minimum, though?

L.132. Change “test set. Calculated the” for “test set. We calculated the”.

L.177. Change “study, the method” for “study, this method”.

L.194. Is “weaken” the correct verb here?

Results

L.215. More accurate than what?

L.220. Table 3: How different or similar were the altitudes among those 30 latitudes? Maybe add this as a Table legend.

L.225. Change “GBRI and VEG” for “GBRI, and VEG”.

L.226. Change “DEVI and g” for “DEVI, and g”.

L.227. Change “RGRI and COM2” for “RGRI, and COM2”.

L.228. Change “g and DEVI” for “g, and DEVI”.

L.229. Change “g and COM” for “g, and COM”.

L.230. Change “RGRI and COM2” for “RGRI, and COM2”.

L.238. Change “RGBVI and EXG” for “RGBVI, and EXG”.

L.239, 241. Change “GBRI and COM” for “GBRI, and COM”.

L.240. Change “EXG and COM2” for “EXG, and COM2”.

L.243. Change “RGBVI and EXG” for “RGBVI, and EXG”.

L.252. Change “g and GBRI” for “g, and GBRI”.

L.253. Change “RGBVI and EXG” for “RGBVI, and EXG”.

L.254. Change “g and GBRI” for “g, and GBRI”.

L.255. Change “EGRBDI and VEG” for “EGRBDI, and VEG”.

L.257. Change “RGBVI and EXG” for “RGBVI, and EXG”.

L.265. Change “CIVE and EXG” for “CIVE, and EXG”.

L.266. Change “DEVI and g” for “DEVI, and g”.

L.267. Change “NGBDI and EXB” for “NGBDI, and EXB”.

L.268. Change “DEVI and g” for “DEVI, and g”.

L.270. Add a comma (,) before “and EXG”.

L.278. Change “EXR and GBRI” for “EXR, and GBRI”.

L.283-284. So, what is the implication of this?

Discussion

L.289. Change “Some previous studies” for “Previous studies”.

L.296. What kind of structural risk do you mean here?

L.299. Change “RGRI and COM2” for “RGRI, and COM2”.

L.300. Change “moderate and slight desertification” for “moderate, and slight desertification”.

L.317. Change “RGBVI and EXG” for “RGBVI, and EXG”.

L.322-323. This part is not completely clear.

L.324. Add a comma (,) after “CIVE and”.

L.327. Change “It may be” for “This may be”.

L.346. Change “texturing and neural network” for “texturing, and neural network”.

Conclusions

L.352. Change “RGRI and COM2 have high accuracy in assessing severe, high, moderate and slight” for “RGRI, and COM2 have high accuracy in assessing severe, high, moderate, and slight”.

Reviewer 2 Report

View letter

The first different is not necessary in the title.

Line 13 can be deleted from the abstract.

Lines 23-25 are not necessary in the abstract.

The results section is not enough in the abstract.

Line 117, the images obtained at different flight heights have different scales and cover different areas. Is there any difference in vegetation index and desertification degree? Such as 10m, 20m, 50 m and so on. I think that the desertification degree is relative to scale bar.

Line 299, the RGRI and COM2 just are suitable for the assessment of severe desertification not suitable for other three types.

Line 317, how about the CIVE.

Line 331, maybe six vegetation indices is accurate.

Line 352, the RGRI and COM2 may not suitable for all grassland desertification.

The content in acknowledge is the same as Funding.

Please carefully check the document suitability, case and other information in the references as, 1, 10, 13, 14, 22, 27, 35, 39, 43, 57, 62 and 70.

Round 2

Reviewer 2 Report

View letter

1.      Is there a standard for grassland desertification, or on what scale to characterize vegetation coverage for grassland desertification ?

2.      Table 3 give the note for OA and k.

3.      Check the language format and other information.
